# Quality Factor Enhancement of Piezoelectric MEMS Resonator Using a Small Cross-Section Connection Phononic Crystal

**DOI:** 10.3390/s22207751

**Published:** 2022-10-12

**Authors:** Lixia Li, Chuang Zhu, Haixia Liu, Yan Li, Qi Wang, Kun Su

**Affiliations:** 1School of Mechanical and Electrical Engineering, Xi’an University of Architecture and Technology, Xi’an 710055, China; 2Institute of Mechanics and Technology, Xi’an University of Architecture and Technology, Xi’an 710055, China

**Keywords:** piezoelectric MEMS resonator, phonon crystals, anchor loss, quality factor, motion resistance, insertion loss

## Abstract

Anchor loss is usually the most significant energy loss factor in Micro-Electro-Mechanical Systems (MEMS) resonators, which seriously hinders the application of MEMS resonators in wireless communication. This paper proposes a cross-section connection phononic crystal (SCC-PnC), which can be used for MEMS resonators of various overtone modes. First, using the finite element method to study the frequency characteristics and delay line of the SCC-PnC band, the SCC-PnC has an ultra-wide bandgap of 56.6–269.6 MHz. Next, the effects of the height *h* and the position *h*_1_ of the structural parameters of the small cross-connected plate on the band gap are studied, and it is found that *h* is more sensitive to the width of the band gap. Further, the SCC-PnC was implanted into the piezoelectric MEMS resonator, and the admittance and insertion loss curves were obtained. The results show that when the arrangement of 4 × 7 SCC-PnC plates is adopted, the anchor quality factors of the third-order overtone, fifth-order overtone, and seventh-order overtone MEMS resonators are increased by 1656 times, 2027 times, and 16 times, respectively.

## 1. Introduction

In recent years, with the rapid development of the fifth generation of mobile communication (5G), MEMS resonators have been widely used due to their small size, low loss, high resonant frequency, and integration with CMOS (Complementary Metal Oxide Semiconductor) circuits. Standard products include MEMS clocks, MEMS accelerometers, MEMS gyroscope, MEMS microphones, etc. [1,2,3,4]. Therefore, it is of great significance to study the method of improving the quality factor of MEMS resonators for these practical applications.

In the field of wireless communication, the RF MEMS devices that are researched and applied by domestic and foreign researchers mainly include MEMS oscillators and MEMS filters. In MEMS oscillator circuits, the higher the quality factor (*Q*) value, the more stable the output frequency and phase; in MEMS filter circuits, the higher the quality factor (*Q*) value, the lower the insertion loss and narrower bandwidth. The core components of both are MEMS resonators, so the design of high-performance resonators is a current hot research topic [5].

The quality factor is a measure of energy loss in a resonator. When energy is converted from one form to another, a part of the energy contained in the system will undergo two forms of conversion. The energy in the system will either escape directly, or repeatedly become a form of energy that cannot be recovered. The quality factor *Q* is usually defined as [6]:(1)$$Q=2\pi \frac{peak\;energy\;stored}{energy\;dissipated\;percycle\;},$$

The above suggests that achieving a higher quality factor requires lower energy dissipation. The common loss of oscillators is divided into inherent loss and external loss. Inherent losses include dielectric loss and piezoelectric loss, and external losses include anchor loss (support loss), surface loss, ohm loss, and so on [6]. Among the many losses, anchor loss is the primary source of energy loss for MEMS resonators, caused by the propagation of acoustic waves from the cavity to the substrate through the support beam [7,8]. The energy carried by this part of the sound waves can not only not be used through the mechanical and electrical conversion, but also may affect the operating performance of other devices. Based on this, it is urgently necessary to use some effective methods to suppress the anchor loss of the resonators. Presently, researchers have proposed a number of methods to reduce the resonator’s anchor loss [6,9,10,11,12,13,14,15,16,17,18,19,20,21,22]. For example, the anchor point is placed [6] at a minor resonator node. The resonator boundary displacement is almost zero; thus, optimizing the loss of the resonator is designed into the butterfly [9], the etching stress release hole [10] on the resonator or the resonator edge to reduce the acoustic energy dissipated through the support beam. Based on the Bragg reflection law, introducing the acoustic mirror [12,13] or the phonon crystal structure [14,15,16,17,18,19,20] can reflect the acoustic waves propagating from the resonator, thus effectively reducing the anchor loss. In recent years, the continuous development of phonon crystal correlation theory will provide a good idea for many scientists to use phonon crystal to reduce the anchor loss of resonators.

Phononic crystals (PnC) are a novel class of periodic synthetic materials that can be used to manipulate the propagation of elastic and acoustic waves [14]. It is well known that well-designed phononic crystal structures provide a remarkable phenomenon known as the acoustic band gap, within which acoustic wave propagation and mechanical vibrations can be prohibited [15]. With the rapid development of phononic crystals in the fields of physics and engineering, many new devices with excellent properties can be fabricated by the rational utilization of phononic crystals. Therefore, the effective use of phononic crystals can suppress the energy leaking through the anchor point, limit the mechanical vibration of the resonator body, and improve the quality factor, thereby helping to improve the resonator’s performance [16].

Due to the significant application of phononic crystals in the field of MEMS resonators, many researchers have been attracted to study in recent years. They reduce anchorage by adding one-dimensional phononic crystals on the support beam, or by using two-dimensional phononic crystals on the substrate point loss [19,20]. In 2018, Lee, J.E.-Y et al. proposed a disc-shaped phononic crystal with a band gap of 93–175 MHz, and the unloaded quality factor *Q* was four times higher than that of ordinary MEMS resonators [16]. In 2019, Bao, F.H et al. proposed a spider web-like PnC to fully isolate acoustic vibrations, with the widest bandgap of 68.0–84.5 MHz and an anchor loss quality factor (*Q**_anchor_*) improved from 5870 to 64,800 [21]. In 2021, Thi Dep Ha proposed a holed circle phononic crystal with a designed resonator operating frequency of around 133 MHz, with a remarkably improved quality factor of the resonator [22]. Since the acoustic band gap formed above is still not wide enough and the application range is relatively limited, it is worth further exploration to design an ultra-wide band gap phononic crystal to improve the resonator’s quality factor.

In this paper, we first introduce an SCC-PnC structure with a very low and ultra-wide acoustic bandgap, and apply SCC-PnC to piezoelectric MEMS resonators to reduce the anchor loss of piezoelectric MEMS resonators. Then, we carried out phononic crystal dispersion curve calculation, delay line simulation, and analysis. Next, the effect of structural parameters on the band gap range of phononic crystals is investigated. Finally, the performance parameters of traditional piezoelectric MEMS resonators and SCC-PnC piezoelectric MEMS resonators are analyzed and compared in three overtones (third, fifth and seventh overtone) vibration modes.

## 2. Materials and Methods

### 2.1. Phononic Crystals Structure

We proposed an SCC-PnC structure. As shown in Figure 1, the structural size of the phonon crystal designed in this paper includes the basic 3-D model of the phonon crystal, (Figure 1b) the top view of the unit structure, (Figure 1c) a partial structure of the phonon crystal, and (Figure 1d) the first Brillouin area of the phonon crystal. The lattice constant of the phonon crystal monocyte is a = 16 µm, the structure height of the square is *H* = 10 µm, the connecting plate *h* = 2 µm, the connecting plate *w* = 1 µm, the central height of the connecting plate is *h*_1_ = 5 µm, and the length is *l* = 2 µm.

Monocrystalline silicon is the most common and widely used single material in MEMS manufacturing. It can be used as a substrate compatible with semiconductor processing equipment and as a structural material for MEMS devices. To exploit this phononic crystal structure in MEMS resonators, the material used in this study is the most commonly used anisotropic single crystal silicon [23,24,25,26]. As shown in Figure 1, in this study, orthotropic monocrystalline silicon was used, and the default *x*-, *y*-, and *z*-axes were set to the (110), (−110), and (001) directions of standard (100) silicon [21]. The specific material parameters used in this simulation are shown in Table 1.

### 2.2. Phononic Crystals Band Gaps

The current methods to calculate the elastic wave band gap include the plane wave expansion method, time domain finite difference method, multiple scattering method, concentrated mass method, finite element method, and others [23,24,25,26]. The finite element method has the advantages of a clear concept, strong applicability, and good convergence. Therefore, this study uses the COMSOL 5.4 Multiphysics finite element analysis system to study the acoustic wave propagation characteristics of phononic crystals. Since the 2D phononic crystal has periodicity in the X and Y directions, by applying the Floquet periodic boundary condition on the unit cell boundary.

Parametric scanning of the first Brillouin region occurs along the Γ-X-M-Γ boundary in the first Brillouin region (IBZ). When *h* = 2 µm, the SCC-PnC dispersion curve is shown in Figure 2a, which has a complete band gap: 56.6–269.6 MHz. Figure 2b shows the mode shape diagram at the start-stop boundary of the SCC-PnC energy band. At point A at 56.6 Mhz, the inner mass is completely used as a vibrator, vibrating along a certain direction of the connecting plate; and at the cut-off frequency point B, it is the torsional vibration of the inner mass. When *h* = *H* = 10 mm, the structure at this time is a traditional phononic crystal (T-PnC), and its dispersion curve is shown in Figure 3a; it also has a complete band gap: 141.8–215.5 MHz. Figure 3b shows the mode shape diagram of the T-PnC band structure at the starting and ending frequencies. At this time, the mode shape of the initial frequency point C is the partial vibration of the internal mass block, and the vibration mass becomes smaller than that of point A. In addition, since the connecting plate acts as a spring at this time, its stiffness increases with *h* and becomes larger, thus resulting in a significant increase in the bandgap onset frequency.
(2)$$u_{i}\left(x+a,y+a\right)=u_{i}\left(x,y\right)e^{\left(j\left(k_{x}a+k_{y}b\right)\right)}\;i=x,y,$$

In Equation (2), where *u* is the displacement field, *k_x_* and *k_y_* represent the wave vector the component in the first Brillouin zone.

In order to better reflect the band gap of phononic crystals, the index of the relative band gap frequency width is introduced in this paper. The relative band gap width *BG*% is defined as the band gap width ratio to the band gap’s center frequency. The solution formula of *BG*% is as follows [22]:(3)$$BG\%=\frac{2(f_{top}-f_{bot})}{f_{top}+f_{bot}},$$

In Equation (3), $f_{top}$ and $f_{bot}$ represent the upper and lower boundary frequencies of the band gap, respectively. High *BG*% represents a wider band gap, and higher *BG*% values indicate better performance of the phonon crystals. As can be seen from parts (a) and (b) of Figure 2, the width of SCC-PnC is about 130.6%, the central frequency (*f_c_*) of band gap is ($f_{top}$ + $f_{bot}$)/2 is 163 MHz; the width of T-PnC is about 41.3%, and the central frequency is 178.65 MHz. The band gap width between the SCC-PnC and the different phonon crystal shapes designed here is compared over similar frequency ranges, as shown in Table 2.

### 2.3. Effect of the Structural Parameters on the Band Gap Range

The phononic crystal’s structural change will affect the band gap’s range. By changing the height *h* of the connecting plate and the position *h*_1_ of the connecting plate, the change law of the starting band gap and the cut-off band gap is studied. As can be seen from parts in Figure 4a,b, keeping other parameters unchanged, with the increase of *h*, the starting frequency of the band gap increases rapidly, while the cut-off frequency of the band gap gradually decreases. With the increase of *h*_1_, the starting frequency of the band gap decreases slowly, while the cut-off frequency of the band gap increases slowly. Therefore, the width of the band gap gradually changes.

### 2.4. Power Transmission Characteristics

Furthermore, to further verify that the relevant PnC structure is able to form a vocal band gap, as shown in Figure 5: Five unit structures are arranged in the *x*-direction and the *y*-direction, respectively. A line displacement excitation signal is applied to one end of the structure, the line displacement response signal is picked up at the other end, and a perfect matching layer (PML) is added to both ends of the structure to eliminate the effect of boundary reflection on the result.

Transmission parameters (i.e., *S*_21_) are introduced in the delay and solid lines to quantify the acoustic isolation effect; *S*_21_ is expressed by decibel (dB) as:(4)$$S_{21}=10log_{10}\left(\frac{P_{out}}{P_{in}}\right)=10log_{10}\left(\frac{d_{out}^{2}}{d_{in}^{2}}\right),$$

In the formula, *P_out_* and *P_in_* are the output power and input power of the excitation surface and the receiving surface, respectively, and *d_out_* and *d_in_* represent the displacement response and displacement excitation, respectively. *S*_21_ is the *S* parameter of the transmitted wave, representing the power transfer coefficient from input port 1 to output port 2.

As shown in Figure 5, by comparing the total displacement amplitude of the phononic crystal delay line and the reference group delay line at 149.66 MHz, we can see that the control group structure cannot prevent the sound wave from passing through the array structure. In contrast, the phononic crystal periodic structure can completely suppress the sound wave within the band gap. By comparing the transmission characteristics of the two groups of phononic crystal delay lines along the Γ-X direction and the control group delay line in Figure 6, the delay line using the phononic crystal has an excellent ability to suppress acoustic waves. At a frequency of 149.66 MHz, the transmission coefficient of the phononic crystal delay line is lower than that of the control delay line, and the acoustic wave suppression capability of the SCC-PnC reduces the acoustic wave transmission coefficient by 156.51 dB, while the T-PnC reduces the acoustic wave transmission coefficient by 77.89 dB. By comparing the above two phononic crystals, the SCC-PnC has a more prominent acoustic wave suppression ability. 

## 3. MEMS Resonator Design and Finite Element Analysis Results

Figure 7 shows a 3-dimensional schematic diagram of the piezoelectric MEMS resonator. In this research, the resonator’s center electrode spacing (Wp) was set at 28.3 µm and designed as a fifth-order symmetric lamb-wave resonator with a resonant frequency of 149.66 MHz. For any given harmonic mode, the resonant frequency of the resonator is provided by the [28,29,30]:(5)$$f=\frac{nv}{2W_{r}},$$
where *v* is the sound speed of the corresponding resonant mode, *n* is the order of the resonant mode, *W_r_* is the width of the resonator, and the spacing between the electrodes is Wp. We set *n* = 5 for the fifth-order width extension mode, *W_r_* = 5 Wp. The specific size parameters of the resonators are shown in Table 3.

This study obtains simulation models for all resonators according to the PML boundary conditions to absorb dissipative sound waves. Calculate the resonator anchor loss (*Q_anchor_*) by [19]:(6)$$Q_{anchor}=\frac{Re\left(f\right)}{2Im\left(f\right)},$$
where *ω* is the resonant frequency of the resonator, *Re*(*ω*) represents the real part of the resonant frequency and *Im*(*ω*) represents the imaginary part of the resonant frequency.

Only a quarter of the simulation model can be established in the simulation to reduce the calculation time, as shown in Figure 8a a traditional type piezoelectric MEMS resonator, and (Figure 8b) a piezoelectric MEMS resonator with a 4 × 7 SCC-PnC array plate. Because the overall structure of the resonator has symmetry, the simulation of the complete model can be equivalent by assigning “symmetrical” boundary conditions on the symmetry surface. Moreover, to avoid the influence of sound wave reflection on the simulation results, the width of the PML is generally set at three times the wavelength so that it can fully absorb the propagating sound waves.

When a potential is applied between the aluminum electrodes, an electric field is then generated in the thickness direction of the piezoelectric film of the resonator. Due to the inverse piezoelectric effect, the applied electric field will cause the piezoelectric film to generate lateral extension stress in the plane so that the resonator vibrates laterally. At specific frequencies, the vibration amplitude of the resonator will reach the maximum and have a regular shape change. The output frequency at this time is called the resonant frequency, and the deformation of the resonator is called the resonant mode. Schematic representation of the total displacement distribution of the MEMS resonators at different overtones is shown in Figure 9. The corresponding eigenfrequencies were analyzed by FEM simulation showing good vibration patterns.

## 4. Discussion

As shown in Figure 9, the FEM simulation results of the traditional piezoelectric MEMS resonator and the phonon crystal piezoelectric MEMS resonator are compared and analyzed. Figure 9 shows the vibration modes of a resonator’s third, fifth, and seventh overtone for both the traditional resonators and the SCC-PnC resonators. The figure shows that the traditional piezoelectric MEMS resonator has a lower total displacement, resulting in a large energy loss. More importantly, under the third-order overtone, the $Q_{anchor}$ result of the phononic crystal resonator is 58,012,627, which is 1656 times higher than the 35,039 of the traditional resonator; under the fifth-order overtone, the $Q_{anchor}$ result of the phononic crystal resonator is 124,052,163, which is 2027 times higher than the 61,207 of the traditional resonator; under the seventh-order overtone, the $Q_{anchor}$ result of the phononic crystal resonator is 462,009, which is 16 times higher than the 28,026 of the traditional resonator.

In addition, the total displacement diagram of the resonant plate at the line A-A’ shown in Figure 10 (see Figure 7 and Figure 8) shows that the displacement of the phononic crystal cavity is the largest. This result indicates that the energy on the phononic crystal resonator is higher than that of the traditional resonator. The phonon crystal can reduce the anchor loss of the resonator more effectively, so that the quality factor of the phonon crystal MEMS resonator is higher.

To further analyze the performance of the designed MEMS resonator, we plot the admittance curves of the resonator under three successive overtones (third-order overtone, fifth-order overtone, and seventh-order overtone), as shown in Figure 11. The following Formulas (7) and (8) can calculate the dynamic resistance and the effective electromechanical coupling coefficient ($K_{eff}^{2}$) of the corresponding resonator:(7)$$R_{m}=\frac{1}{\max\left\{ReY11\right\}},$$
(8)$$K_{eff}^{2}=\frac{f_{p}^{2}-f_{s}^{2}}{f_{p}^{2}},$$
where max {*Re*(*Y*11)} is the maximum real part of admittance, *f_p_* is the frequency at which the impedance amplitude is maximum, and *f*_s_ is the frequency when the impedance amplitude is the minimum.

Figure 12 shows a comparison chart of the insertion loss (*S*_21_) curves of three successive overtones (third-order overtone, fifth-order overtone, and seventh-order overtone). Under three consecutive overtones, the insertion loss (IL) of the piezoelectric MEMS resonator is reduced from 10.54 dB, 4.18 dB, and 15.13 dB to 1.02 dB, 1.87 dB, and 3.28. Since the simulation is in the no-load process, the on-load quality factor (*Q_l_*_)_ can be obtained by calculating the 3 dB bandwidth by Equation (9), and the unloaded quality factor (*Q_u_*) can be obtained according to Equation (10). We can see from the *S*_21_ curve obtained from the simulation that after the SCC-PnC array is integrated into the resonator, the equivalent density of the resonator changes, which will cause the resonant frequency to change. Still, the change is minimal, and the change of the resonant frequency can be controlled by adjusting the structure size. The above analysis and calculation show how the specific performance parameters of the traditional resonator and the SCC-PnC resonator under different overtones can be compared. The specific performance parameters are shown in Table 4.
(9)$$Q_{l}=\frac{f_{s}}{\Delta f_{-3\mathrm{dB}}},$$
(10)$$Q_{u}=\frac{Q_{l}}{1-10^{-\frac{IL}{20}}},$$
where $\Delta f_{-3dB}$ is −3 dB band width and *IL* is insertion loss. 

## 5. Conclusions

This work provides an SCC-PnC structure, using finite element simulation methods to systematically analyze the PnC’s band structure and transport properties. Compared with a PnC structure in a similar vocal bandgap frequency range, the proposed SCC-PnC has a wider band gap (56.6–269.6 MHz) and a larger relative band gap frequency width of *BG*% (130.6%). Next, the influence of the height *h* and position *h*1 of the structural parameters of the small cross-connecting plate on the band gap is studied. It is found that *h* is more sensitive to the width of the band gap. When SCC-PnC is injected into the piezoelectric MEMS resonator, there is no significant effect on the effective electromechanical coupling. In three continuous overtone modes (third overtone, fifth overtone, seventh overtone), *Q_l_* is increased from 25,068, 16,629, 265,13 to 85,948, 21,079, and 103,585, respectively. In these three continuous overtone modes, the motional resistance is reduced from 235.29 Ω, 18.18 Ω, and 156.25 Ω to 3.39 Ω, 4.29 Ω, and 48.31 Ω, respectively. Meanwhile, under these three consecutive overtone modes, the insertion loss of the SCC-PnC resonator is reduced by 9.52 dB, 2.25 dB, and 11.85 dB, respectively, compared with the traditional resonator. This study provides a good idea for the follow-up research on improving the performance of piezoelectric MEMS resonators.

## Figures and Tables

**Figure 1 sensors-22-07751-f001:**
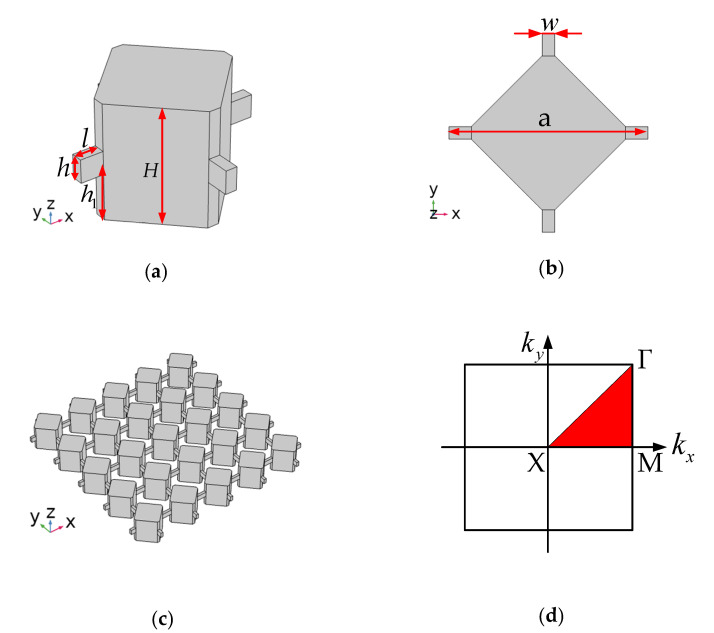
Phononic crystal structure size display: (**a**) the basic 3D model of the phonon crystal; (**b**) the top view of the unit structure; (**c**) a partial structure of the phonon crystal; (**d**) the first Brillouin area of the phonon crystal.

**Figure 2 sensors-22-07751-f002:**
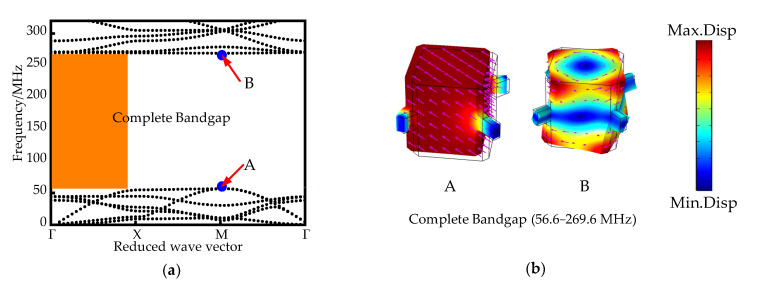
(**a**) SCC-PnC dispersion curve (**b**) Vibration mode diagram at the start and end points of the band gap (The arrow represents the size and direction of the displacement).

**Figure 3 sensors-22-07751-f003:**
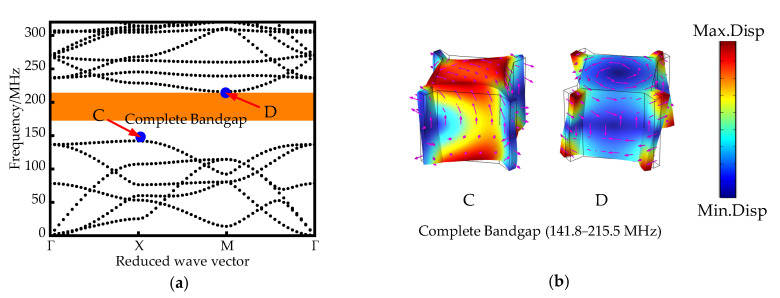
(**a**) Traditional PnC dispersion curve (**b**) Vibration mode diagram of the start and end points of the band gap (The arrow represents the size and direction of the displacement).

**Figure 4 sensors-22-07751-f004:**
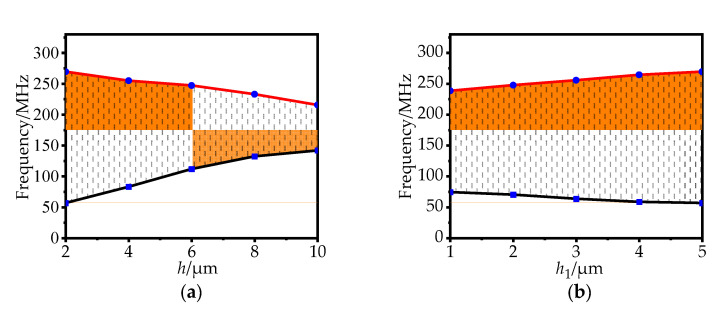
(**a**) The effect of *h* on the band gap; (**b**) The effect of *h*_1_ on the band gap.

**Figure 5 sensors-22-07751-f005:**
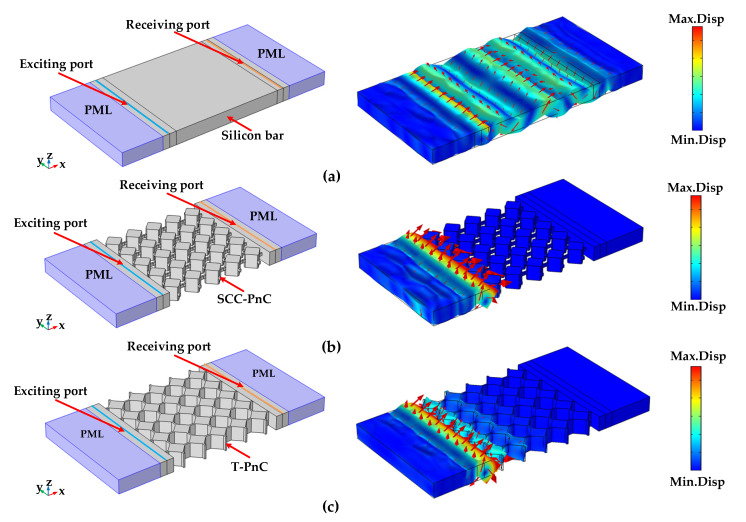
Comparison of total displacement amplitudes of (**a**) reference group delay line, (**b**) SCC-PnC delay line, and (**c**) T-PnC delay line at 149.66 MHz (The arrow represents the size and direction of the displacement).

**Figure 6 sensors-22-07751-f006:**
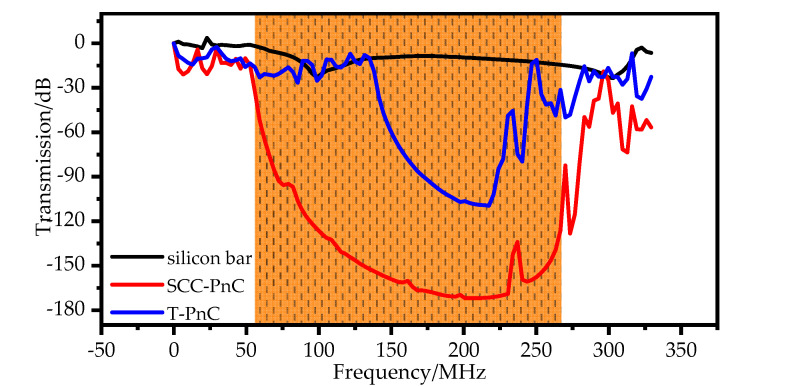
Transmission spectra of two kinds of phononic crystals and reference group along the Γ−X direction.

**Figure 7 sensors-22-07751-f007:**
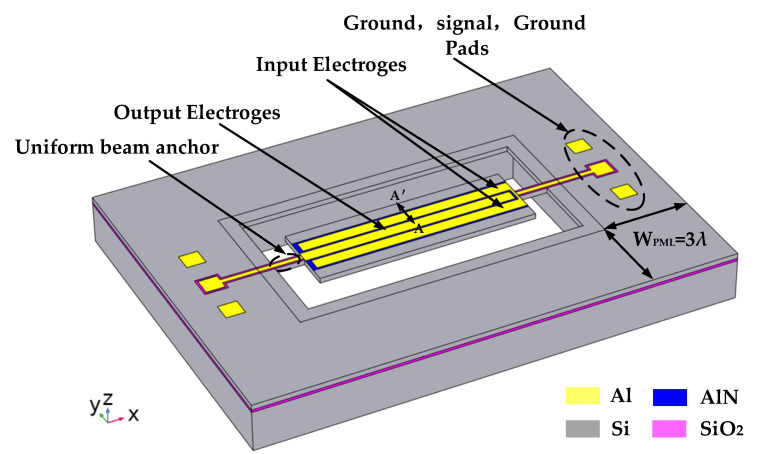
Model diagram of the piezoelectric MEMS resonator designed in this research.

**Figure 8 sensors-22-07751-f008:**
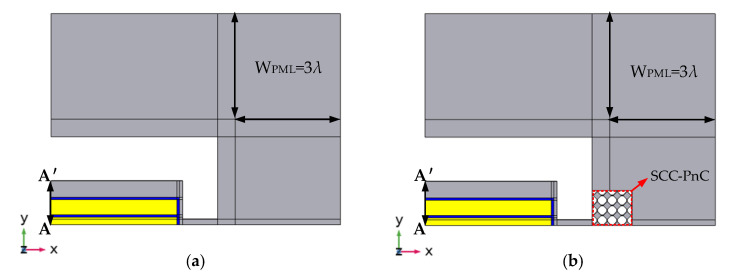
Simulation diagrams of quarter piezoelectric MEMS resonators: (**a**) traditional type piezoelectric MEMS resonator; (**b**) piezoelectric MEMS resonator with 4 × 7 SCC-PnC array plate.

**Figure 9 sensors-22-07751-f009:**
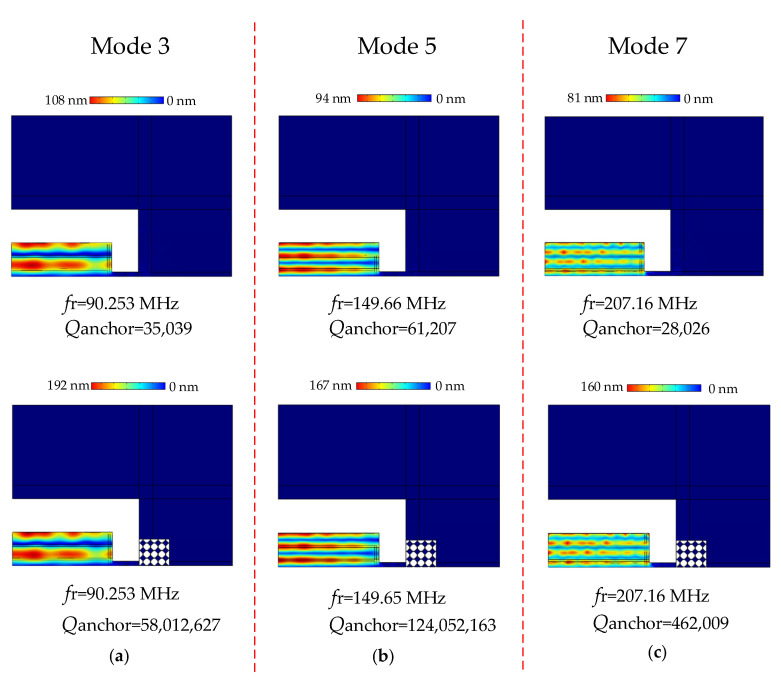
Schematic diagram of total displacement distribution of different MEMS resonators: Finite element simulation of vibration modes and quality factors for the (**a**) third (90.25 MHz, Mode 3), (**b**) fifth (149.66 MHz, Mode 5), and (**c**) seventh (207.16 MHz, Mode 7) overtone resonators of conventional resonators and SCC-PnC resonators.

**Figure 10 sensors-22-07751-f010:**
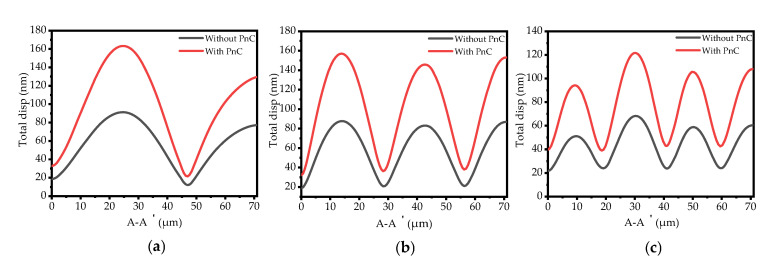
Illustration of the total displacement (μm) for the conventional resonator and the resonator with the SCC-PnC plate for the (**a**) 3rd, (**b**) 5th, and (**c**) 7th order width-extended modes on line A-A, respectively.

**Figure 11 sensors-22-07751-f011:**
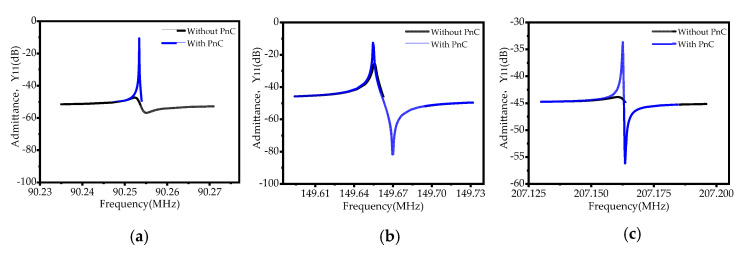
Admittance curves of the resonator under three successive overtones ((**a**) third−order over-tone, (**b**) fifth−order overtone, and (**c**) seventh−order overtone).

**Figure 12 sensors-22-07751-f012:**
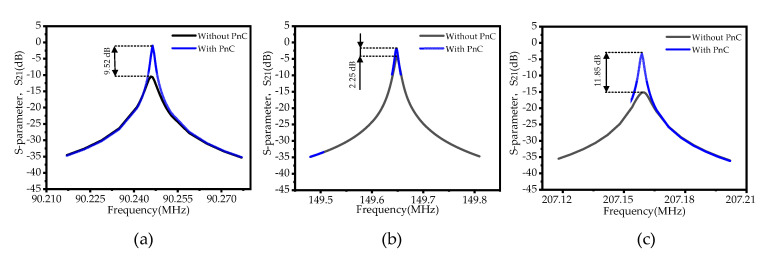
Comparison of insertion loss (*S*_21_) curves of three consecutive overtones ((**a**) third−order overtone, (**b**) fifth−order overtone, and (**c**) seventh−order overtone).

**Table 1 sensors-22-07751-t001:** The specific material parameters used in this simulation.

Material	Parameters	Values
Silicon (Si)	Mass Density (kg/m^3^)	2330
	Young’s Modulus (GPa)	$E_{x}\;$= 169
		$E_{y}$ = 169
		$E_{z}$ = 130
	shearing’s Modulus (GPa)	$G_{yz}$ = 79.6
		$G_{zx}$ = 79.6
		$G_{xy}$ = 50.9
	Poisson’s ratio (ν)	$\sigma_{yz}$ = 0.36
		$\sigma_{zx}$ = 0.28
		$\sigma_{xy}$ = 0.064
Alµminµm (Al)	Mass Density (kg/m^3^)	2700
	Young’s Modulus (GPa)	70
	Poisson’s ratio (ν)	0.35
	Electrical conductivity (σ)	33.5 × 10^6^
	Coefficient of thermal expansion (α)	23.1 × 10^−6^
	Heat capacity (Cp)	904
	Thermal conductivity (κ)	237
Alµminµm Nitride (AIN)	Mass Density (kg/m^3^)	3300
	Young’s Modulus (GPa)	320
	Poisson’s ratio (ν)	0.24
	Relative permittivity (ε)	9

**Table 2 sensors-22-07751-t002:** Comparison of simulated *BG%* between various PnC shapes using a similar lattice parameter.

PnC Shape	Dimensions (µm)	Range (MHz)	*f_c_* (MHz)	*BG*%
Air-Hole [16]	22	136–147	141.5	7.7
DTP [18]	16	105–240	173	78
Solid-Disk [27]	22	93–175	134	61.1
SCC (This work)	16	56.6–269.6	163	130.6

**Table 3 sensors-22-07751-t003:** The specific size parameters of the resonators.

Parameters	Values (Unit)
Simulated resonant frequency (*f*_0_)	149.66 (MHz)
Wave length (*λ*)	56.6 (µm)
Inter digitated transducer (IDT) finger (*n*)	5
Tethers Width (*W**_t_*)	18 (µm)
Tethers Length (*L**_t_*)	56.6 (µm)
electrode gap (*G**_e_*)	4 (µm)
Resonator width (*W**_r_*)	141.5 (µm)
Resonator length (*L**_r_*)	424.5 (µm)
Thickness of Al (*T**_Al_*)	0.1 (µm)
Thickness of AlN (*T**_AlN_*)	0.1 (µm)
Height of silicon substrate (*H**_S_*)	10 (µm)

**Table 4 sensors-22-07751-t004:** Comparison of performance parameters of four MEMS resonators.

Parameters	Mode 3	Mode 3 (with PnC)	Mode 5	Mode 5 (with PnC)	Mode 7	Mode 7 (with PnC)
Resonant frequency (*f*_r_), MHz	90.25	90.25	149.66	149.65	207.16	207.16
Insertion Loss (*IL*), dB	10.54	1.02	4.18	1.87	15.13	3.28
Motional resistance (*R**_m_*), Ω	235.29	3.39	18.18	4.29	156.25	48.31
Coupling coefficient (*K**_eff_*^2^), %	0.006	0.006	0.02	0.02	0.006	0.006
Simulated *Q**_anchor_*	35,039	58,012,627	61,207	124,051,263	28,026	462,009
Loaded Quaity factor (*Q_l_*)	25,068	85,948	16,629	21,079	26,513	103,585
Unloaded Quality factor (*Q_u_*)	35,659	79,0326	43,531	108,823	32,137	329,259

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
