# Peer review of "Quality Factor Enhancement of Piezoelectric MEMS Resonator Using a Small Cross-Section Connection Phononic Crystal"

_sensors, 2022, doi:10.3390/s22207751_

Round 1

Reviewer 1 Report

Review of the manuscript sensors-1942517, by Li et al.

I read this manuscript with interest and find it interesting for application purposes. Certainly, the introductory topics should be supplemented with a more recent bibliography including for example Sensors 2021, 21(5), 1871; https://doi.org/10.3390/s2105187.  

The manuscript certainly has novel elements that can be appreciated by an audience in the MEMS framework. In any case, I highlight some unclear parts:

 1) The authors should better summarise and clarify the aspects related to the functional Stress-strain relationship of the composite.

2) The relationship 6) on page 8 not well defined, what do Re(f) and Im(f) define? The real and imaginary part of what?

3) Same clarification for the relationships 7), 9), and 10). They need to be better defined.

4) The conclusions are too reductive compared to the discussion expressed in the previous paragraphs.

I invite the authors to better clarify some expressions in English. I believe that, after these minor revisions, the manuscript can be accepted in Sensors Journal.

Reviewer 2 Report

The author's work provides an SCC-PnC structure, using finite element simulation methods to analyze the PnC's band structure and transport properties systematically.

The work is very good and interesting; they should improve it in the conclusions because it does not reflect the good results obtained and presented in the manuscript.

Reviewer 3 Report

The article addresses the issue of MEMS piezoceramic resonators. In the framework of the undertaken study, both aspects related to the improvement of their characteristics, but also elements of finite modelling and determination of the sensitivity of the studied resonator are addressed. In my opinion, the fort point of the manuscript is the finite element method.

The paper is suitable for this journal, but in my opinion, it must be improved in some parts of the study before being accepted.

1. From the point of view of the Abstract, the structure of this respects the recommendations of the Template proposed to be addressed by the authors of the study. The only aspect that was not respected is the length of the text, which is 211 words. Perhaps it would be recommended that some of the aspects presented be reduced as much as the information presented.

2. From the point of view of the Introduction, there are some aspects of spelling that should be corrected, see for example Line 37 (;) instead of (,). In this part also, I think that a detailed description of the mentioned bibliographic elements is required, with the mention of their importance in relation to the study undertaken (as an example Line 33). The explanation related to equation 1 for the Q factor should be corrected because it does not respect the link between the bibliographic source and the number of equation Lines 44 to 46.

3. From the point of view of Chapter 2, this title must be modified in accordance with the Template (Materials and Methods). An aspect that must be considered and corrected is the way of referencing the figures in the text as an example Line 101 (Figure 1d without putting brackets as recommended in the Template). It should also be detailed, for example, the reference to the bibliographic sources in Line 115 as an example. Also, the connection between the figures and the text should be introduced in a natural order, the reference in the text before the figure as an example Figure 2 or equation 2. The aspect mentioned in the Introduction related to the reference of the equations is repeated in this paragraph Lines 160 to 161. It would be interesting for readers to see a presentation of the resonator from a physical point of view as is presented in this chapter.

4. Chapter 3 must be one of results as the title. I suggest rethinking its part to agree with the imposed model. It must be pointed out that the same types of corrections related to the reference of figures and equations are imposed in this chapter as well.

5. Given the required model of writing such a study, the last part must be corrected. (References).
